cellular biology/molecular biology

astrocyte, human, insulin, lipid storage

**Author for correspondence:**
Martin Heni
e-mail: martin.heni@med.uni-tuebingen.de

# Ectopic fat accumulation in human astrocytes impairs insulin action

Martin Heni[1,2,3,4], Sabine S. Eckstein[2,3],
Jens Schittenhelm[5], Anja Böhm[1,2,3], Norbert Hogrefe[9],
Martin Irmler[6], Johannes Beckers[3,6,7],
Martin Hrabě de Angelis[3,6,7], Hans-Ulrich Häring[1,2,3],
Andreas Fritsche[1,2,3] and Harald Staiger[2,3,8]

[1]Department of Internal Medicine, Division of Endocrinology, Diabetology, and Nephrology, Eberhard Karls University Tübingen, Tübingen, Germany
[2]Institute for Diabetes Research and Metabolic Diseases of the Helmholtz Center Munich at the University of Tübingen, Tübingen, Germany
[3]German Center for Diabetes Research (DZD), Neuherberg, Germany
[4]Department for Diagnostic Laboratory Medicine, Institute for Clinical Chemistry and Pathobiochemistry, and [5]Division of Neuropathology, University Hospital Tübingen, Tübingen, Germany
[6]Institute of Experimental Genetics, Helmholtz Zentrum München, German Research Center for Environmental Health (GmbH), Neuherberg, Germany
[7]Chair for Experimental Genetics, Technische Universität München, Freising, Germany
[8]Institute of Pharmaceutical Sciences, Department of Pharmacy and Biochemistry, Eberhard Karls University Tübingen, Tübingen, Germany
[9]Department of Physiology, University of Bern, Bern, Switzerland

 MH, 0000-0002-8462-3832; SSE, 0000-0001-8063-4465;
MHdA, 0000-0002-7898-2353

Astrocytes provide neurons with structural support and energy in form of lactate, modulate synaptic transmission, are insulin sensitive and act as gatekeeper for water, ions, glutamate and second messengers. Furthermore, astrocytes are important for glucose sensing, possess neuroendocrine functions and also play an important role in cerebral lipid metabolism. To answer the question, if there is a connection between lipid metabolism and insulin action in human astrocytes, we investigated if storage of ectopic lipids in human astrocytes has an impact on insulin signalling in those cells. Human astrocytes were cultured in the presence of a lipid emulsion, consisting of fatty acids and triglycerides, to induce ectopic lipid storage. After several days, cells were stimulated with insulin and gene expression profiling was performed. In addition, phosphorylation of Akt as well as glycogen synthesis and cell proliferation was assessed. Ectopic lipid storage was detected in human astrocytes after lipid exposure

and lipid storage was persistent even when the fat emulsion was removed from the cell culture medium. Chronic exposure to lipids induced profound changes in the gene expression profile, whereby some genes showed a reversible gene expression profile upon removal of fat, and some did not. This included FOXO-dependent expression patterns. Furthermore, insulin-induced phosphorylation of Akt was diminished and also insulin-induced glycogen synthesis and proliferation was impaired in lipid-laden astrocytes. Chronic lipid exposure induces lipid storage in human astrocytes accompanied by insulin resistance. Analyses of the gene expression pattern indicated the potential of a partially reversible gene expression profile. Targeting astrocytic insulin resistance by reducing ectopic lipid load might represent a promising treatment target for insulin resistance of the brain in obesity, diabetes and neurodegeneration.

## 1. Introduction

Studies have not only revealed the existence of insulin receptors throughout the brain [1,2] but also shown that insulin has an important role in various brain functions such as control of body weight, food intake [3] and memory formation [4]. Similar to the periphery, the central nervous system is sensitive to insulin in healthy humans, but can become insulin resistant in a number of conditions, including obesity, type 2 diabetes and neurodegenerative diseases [5]. Brain insulin resistance is unfavourable, as brain insulin resistant individuals show a reduced capacity for weight loss during a lifestyle intervention program [6]. Furthermore, central nervous insulin resistance in overweight people leads to blunted postprandial satiety signals [7,8] and brain-derived regulation of peripheral insulin sensitivity is impaired in obese patients [9,10]. Human studies from our department have shown that the brain response to insulin is diminished in the presence of elevated saturated non-esterified fatty acids [11]. In addition, brain insulin action was reduced in prefrontal cortex and hypothalamus of obese men, leading to an altered homeostatic set point and reduced inhibitory control, contributing to overeating behaviour [12].

Non-neuronal cells outnumber neuronal cells in the central nervous system. This population of non-neuronal cells comprises astrocytes, oligodendrocytes, microglia, ependymal and—in the pituitary gland—specialized epithelial cells, whereby astrocytes represent the most abundant non-neuronal cell type in the brain. Besides providing structural support, astrocytes supply neuronal cells with lactate and amino acids and act as gatekeeper for water, ions, glutamate and second messengers [13]. They furthermore modulate synaptic transmission and are part of the blood–brain barrier [14,15]. In addition, research over the last decades revealed that astrocytes have neuroendocrine functions and it has become clear that astrocytes are part of metabolic circuits within the brain [16]. Glucose uptake and storage in the form of glycogen ensures energy supply for neurons during intense periods of activation or during hypoglycaemia [17]. Lactate as a product of glycolysis in astrocytes is used by neurons as energy source. Furthermore, lactate shuttling from astrocytes towards neurons is crucial for higher cognitive brain functions [18,19]. We have previously shown that astrocytes are sensitive to insulin and that insulin promotes glycogen storage and cell proliferation in astrocytes [20]. García-Cáceres et al. reported an important role of hypothalamic astrocytes in glucose sensing [21]. The authors show that insulin signalling in hypothalamic astrocytes regulates glucose sensing and systemic metabolism by controlling glucose uptake into the brain [21]. Furthermore, loss of astrocytic insulin signalling in mice results in anxiety and depressive-like behaviour [22]. Importantly, these cells appear to be crucial for the brain-derived control of whole-body metabolism [23].

Astrocytes and oligodendrocytes in grey and white matter also play an important role in cerebral lipid metabolism [24] and astrocytes present the main provider of fatty acid-related β-oxidation in the brain [25,26]. Hofmann et al. incubated mouse brain slice cultures with traceable alkyne analogues of fatty acids and observed fatty acid uptake into astrocytes and oligodendrocytes [24]. In further experiments in cultures of mouse primary astrocytes, they observed fatty acid uptake and biosynthesis of membrane lipids and neutral lipids such as cholesterol ester, di- and triacylglycerol. The ability of astrocytes to synthesize cholesterol is vital for normal brain development, body composition and metabolism [27]. Gao et al. showed in a series of experiments in mice that fatty acid uptake mediated by lipoprotein lipase (LPL) in astrocytes is essential for the control of cellular lipid storage [28]. Disruption of this lipid uptake in astrocytes in rodents exacerbates high fat diet-induced body weight gain and glucose intolerance in obesity [28]. The authors speculate that ceramides accumulate as a consequence of disrupted lipid storage, contributing to the pathomechanisms in obesity. Kwon et al.

showed formation of lipid droplets in mouse astrocytes and observed an inflammatory response from lipid-laden astrocytes [29].

The extent of astrocytes' contribution to brain insulin effects in humans is still unclear. Astrocytes are sensitive to insulin and play an important role in lipid and glucose metabolism. This raises the question whether there is a connection between lipid metabolism and insulin sensitivity and action in human astrocytes and what might be underlying mechanisms. We, therefore, investigated if storage of ectopic lipids in human astrocytes has an impact on insulin signalling in those cells.

# 2. Material and methods

## 2.1. Materials

Human insulin (Insuman Rapid) was from Sanofi (Frankfurt, Germany) and Lipofundin® MCT 20% from B. Braun Melsungen (Melsungen, Germany). $^{14}C$-D-glucose was from GE Healthcare (Little Chalfont, UK).

## 2.2. Cell culture

Normal human astrocytes (CC-2565) were purchased from Lonza and grown in AGM medium consisting of astrocyte basal medium (CC-3187, Lonza), 3% fetal bovine serum (FBS) and supplements (ascorbic acid, rhEGF, gentamycin, glutamine) from SingleQuot Kit Suppl. & Growth Factors (CC-4123, Lonza). Cells were grown for 24 h in AGM medium prior to any treatment. For ectopic fat storage, cells were grown for 4–7 consecutive days in AGM medium supplemented with varying Lipofundin concentrations (0.5, 0.05 or 0.005%) and medium was refreshed every 2 days. Every time medium was refreshed, dilutions for the desired Lipofundin concentrations were also freshly prepared. If not otherwise indicated, cells were starved for 24 h in DMEM containing 0.5% FBS, $1 \, g \, l^{-1}$ glucose, 1% penicillin/streptomycin and 1% glutamine before stimulation with 50 nM insulin. Lonza holds donor consent and legal authorization that provides permission for usage of normal human astrocytes for research purposes. Cell are routinely characterized by Lonza with immunofluorescence staining and morphological observation throughout serial passages and staining for glial fibrillary acid protein (GFAP). Only batches > 80% positive for GFAP are purchasable.

## 2.3. Western blot

Cells were lysed in lysis buffer (50 mM Tris, 200 mM NaCl, 100 mM NaF, 5 mM EDTA, 1% Triton X-100, 10% glycerol, 1 mM phenylmethylsulfonylfluoride, 10 µg ml$^{-1}$ aprotinin, 1 mM sodium orthovanadate, pH 7.4) and cleared by centrifugation with $13\,000g$ for 15 min at 4°C. Protein concentration was determined using Bradford Assay (Bio Rad), and equal amounts of protein were separated by SDS-PAGE (sodium dodecyl sulfate polyacrylamide gel electrophoresis). Proteins were transferred onto nitrocellulose membranes (Schleicher & Schuell) and after blocking of unspecific binding sites, incubated overnight with primary antibody. Three times washing followed incubation with horseradish peroxidase-conjugated secondary antibody and three subsequent washes thereafter were followed by detection with ECL. The membrane with p-Ser-473 Akt antibody was stripped and reprobed with Akt protein antibody or Akt protein was detected from a parallel gel. Used antibodies: p-Ser-473 Akt, rabbit (Cat. No. 9271, Cell Signalling); Akt protein antibody (Cat. No. 2920, Cell Signalling).

## 2.4. Oil Red O staining

For Oil Red O staining, astrocytes were cultured for 7 days with either 0.05% or 0.005% Lipofundin and a subset of cells was additionally cultured for 3 days without Lipofundin. After fixation of cells using 10% paraformaldehyde, lipids were stained with Oil Red O (Sigma Aldrich), whereas nuclei were stained with Mayer's hemalum solution (Merck). A Nikon Microscope ECLIPSE 80i with 20× objective was used for image acquisition. Cells were seeded in parallel for the different time points, therefore images represent separate cultures.

**Table 1.** Experimental conditions used for gene expression studies in human astrocytes.

| label | treatment |
| --- | --- |
| 1 | control (con) |
| 2 | 3 h insulin (ins) |
| 3 | 4 days Lipofundin (lipids) |
| 4 | 4 days Lipofundin; 3 h insulin (lipids → ins) |
| 5 | 4 days Lipofundin; 3 days Lipofundin withdrawal (lipids → no lipids) |
| 6 | 4 days Lipofundin; 3 days Lipofundin withdrawal; 3 h insulin (lipids → no lipids → ins) |

## 2.5. Measurement of lactate dehydrogenase

Cells were cultivated for 5 days with 0.5, 0.05 or 0.005% Lipofundin. Cell culture supernatant was removed, and after washing with PBS, cells were lysed with 500 µl $H_2O$ and centrifuged with 13 000$g$ for 10 min. LDH concentration was immediately measured in cell culture supernatant and lysate with a Siemens ADVIA 1800 Clinical Chemistry Analyzer.

## 2.6. Measurement of glycogen synthesis

Cells were grown in plain AGM medium or supplemented with 0.05 or 0.005% Lipofundin for 5 days. Cells were starved with DMEM containing 0.5% FBS, 1 g l$^{-1}$ glucose, 1% penicillin/streptomycin, 1% glutamine and Lipofundin for 24 h prior to stimulation with 50 nM insulin in DMEM medium containing $^{14}$C-D-glucose (0.6 µCi/well) for 3 h. Following stimulation, supernatant was discarded and cells were washed three times with ice-cold PBS. Then 500 µl KOH (30%) was added to cells for 30 min at room temperature. Cell lysate was mixed with 1 mg of glycogen, boiled for 30 min at 95°C and washed twice with ice-cold ethanol and ultimately resuspended in $H_2O$ and analysed with a beta-counter.

## 2.7. Cell proliferation assay

Cells were grown in 96-well format in AGM medium supplemented with 0.05 or 0.005% Lipofundin for 5 days before starvation for 24 h and subsequent stimulation for 3 additional days with 50 nM insulin and respective Lipofundin concentrations. Water soluble tetrazolium (WST)-1 assay (Roche) was used according to the manufacturer's instructions, and absorbance was measured using a Tecan Sunrise™ microplate reader.

## 2.8. Gene expression profiling

Human astrocytes were grown in several conditions (table 1 and figure 3$a$). RNA was isolated using RNeasy Mini Kit (Cat. No. 217004, Qiagen), and 1 µg RNA was transcribed to cDNA with the Transcriptor First Strand cDNA Synthesis Kit from Roche (Cat. No. 04 897 030 001). Gene expression profiling was performed using Human Genome U219 Array Plate. Annotation was downloaded manually from Affymetrix (NetAffx, November 2014). Statistical analyses was performed using R [30] implemented in CARMAweb [31]. Genewise testing for differential expression was done employing the limma $t$-test ($p < 0.05$) and regulated genes were filtered for fold-change > 1.3× and average expression in at least one group in the dataset greater than 10. Fat reversible genes were selected by filtering the 431 gene set (lipids versus con) for genes with regulation in the same direction in the 'lipids → no lipids versus con' condition. To obtain the set of 535 genes regulated by insulin in the absence versus presence of lipids, we compared the ratios of both conditions. Ratios were obtained by dividing values of the insulin treatments by the average of the corresponding control group. To control for outliers, genes with a relative standard deviation greater than 0.2 were excluded from the results. Genes were selected by a limma $p$-value < 0.01, fold-change greater than 1.4× and average expression in at least one group in the dataset greater than 10. Heatmaps were done in R. Pathway analyses were generated through the use of QIAGEN's Ingenuity Pathway Analysis software (IPA®, QIAGEN Redwood City, www.qiagen.com/ingenuity) using Fishers' Exact Test $p$-values.

## 2.9. Statistical analyses

Datasets are presented as mean ± s.e.m. and were tested with one-way ANOVA to detect significant differences between groups. *indicates a significant difference in a *post hoc* test within groups. A result was considered statistically significant when $p < 0.05$. All probe sets of the gene expression data were tested with limma *t*-test (filter: $p < 0.05$). Data were filtered afterwards for average expression of greater than 10 in at least one of the analysed groups and fold change of greater than 1.3 (up or down).

# 3. Results

## 3.1. Human astrocytes form lipid droplets when exposed to fatty acids

Lipofundin is a fat emulsion for intravenous infusion in human therapy and is composed of soya bean extract, medium-chain triglycerides and essential fatty acids. It contains medium-chain fatty acids as triglycerides, long-chain triglycerides from the soya bean extract and the essential free fatty acids linoleic acid and α-linoleic acid, and is usually used as part of a full parenteral diet. To determine working concentrations for Lipofundin treatment that did not exert negative effects on cell viability, astrocytes were cultured with varying Lipofundin concentrations, and release of lactate dehydrogenase (LDH) into the medium was measured as marker for cell viability. High Lipofundin concentration of 0.5% led to significant release of LDH into medium, whereas 0.05 and 0.005% Lipofundin did not (figure 1*a*). Thus, subsequent experiments were performed with 0.005 and 0.05% Lipofundin. Of note, triglyceride concentration was 10 mg dl$^{-1}$ for 0.05% Lipofundin and 1 mg dl$^{-1}$ for 0.005% Lipofundin, whereas linoleic acid had a concentration of approximately 100 µM for 0.05% Lipofundin and approximately 10 µM for 0.005% Lipofundin. α-linoleic acid ranged between 9–18 µM for 0.05% Lipofundin and between 0.9–1.8 µM for 0.005% Lipofundin.

Astrocytes were cultured with Lipofundin for 7 days, and Oil Red O stains showed storage of fatty acids as mostly non-confluent, fine lipid droplets already after 3 days in two independent experiments (figure 1*b,c*). These fatty acid storage compartments were preserved when astrocytes were cultured for another 3 days without Lipofundin. It is assumed that lipid droplets form discrete regions at the endoplasmic reticulum, since the lipid monolayer has a similar phospholipid composition as the endoplasmic reticulum [32]. In line, the majority of the lipid droplets that were observed in our experiments were located around the nucleus.

## 3.2. Ectopic fat storage in astrocytes impairs insulin-dependent activation of Akt, glycogen synthesis and cell proliferation

Human astrocytes were treated with either 0.05 or 0.005% Lipofundin for 5 days to induce ectopic fat storage. To distinguish further between effects mediated by ectopic fat storage in lipid droplets and effects that are introduced by the fatty acids in the supernatant, a subset of cells was cultured for 3 more days without Lipofundin. Cells were then incubated with 50 nM insulin for 30 min to activate the insulin signalling cascade. Pretreatment of astrocytes with 0.005% Lipofundin prior to insulin stimulation did not lead to reduced Ser-473 Akt phosphorylation compared to the control condition (figure 2*a*). However, incubation of cells with the higher Lipofundin concentration of 0.05% led to significantly reduced phosphorylation of Ser-473 Akt in comparison to cells grown without Lipofundin (figure 2*b*). This impaired Akt phosphorylation was still present when cells were cultivated for 5 days in the presence of Lipofundin and subsequently for 3 more days without Lipofundin prior to insulin stimulation. Thus, ectopic fat accumulation, but not acute fatty acid signalling, in human astrocytes impairs activation of the insulin signalling cascade.

As to the more functional assessments, insulin-stimulated glycogen synthesis was also blunted in astrocytes loaded with fatty acids at both tested Lipofundin concentrations (figure 2*c*). In a cell proliferation assay, we observed an increase in cell number when astrocytes were solely incubated with 0.05% Lipofundin, but diminished insulin-induced proliferation of human astrocytes when challenged with both Lipofundin concentrations for 7 days (figure 2*d*).

## 3.3. Lipofundin treatment alters gene expression profile in human astrocytes

In a pilot experiment, the different effects of insulin and lipids on the gene expression profile in human astrocytes were evaluated. The following questions were drafted: (i) what is the effect of insulin on gene

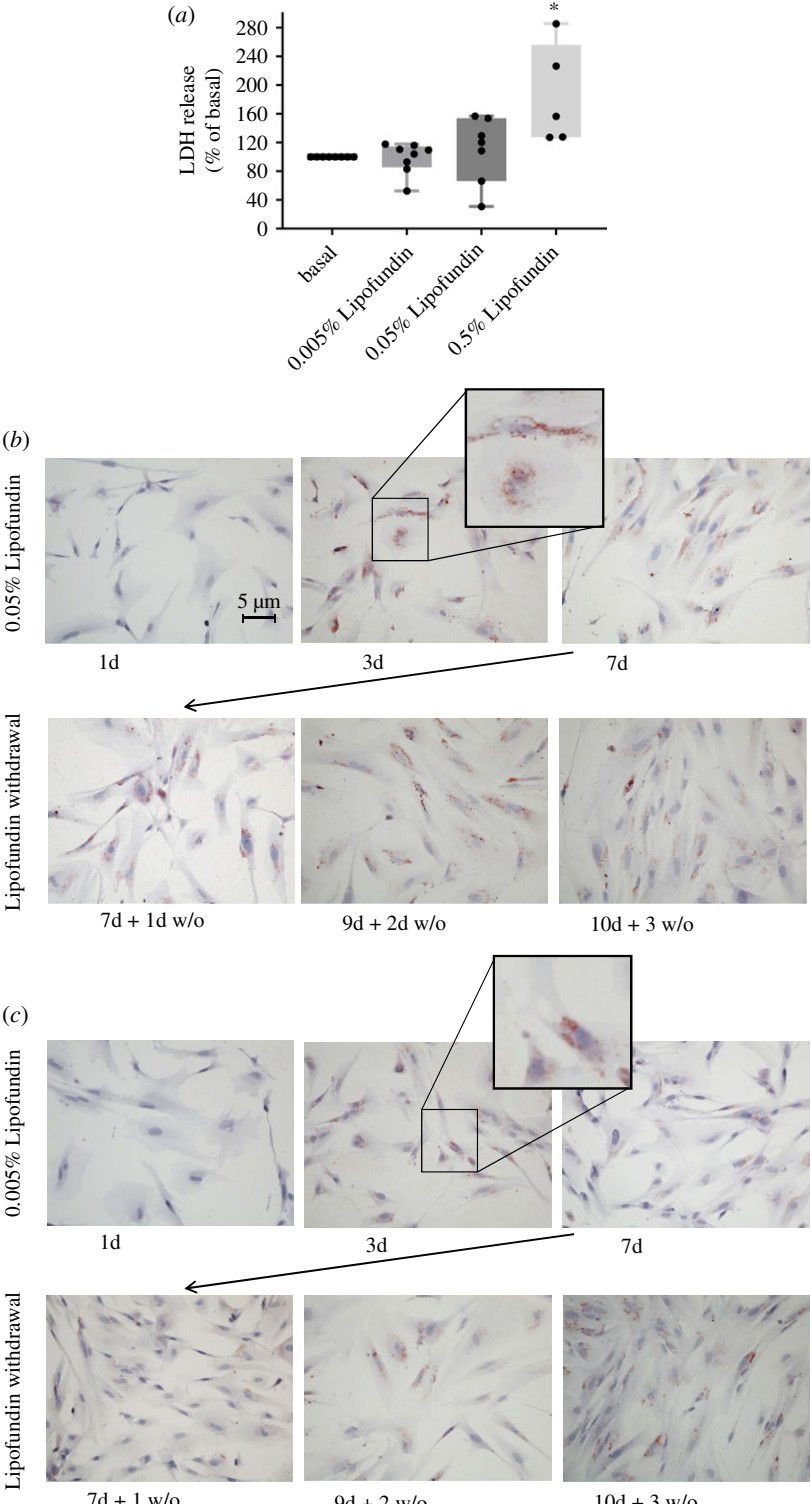

**Figure 1.** Effect of Lipofundin on cell viability and lipid droplet formation in human astrocytes. (*a*) LDH release in human astrocytes. Cells were treated with the indicated Lipofundin concentration for 5 days. Thereafter, LDH was determined in supernatants and cell lysates. LDH was analysed in relation to intracellular LDH content for each condition. Cells cultured without Lipofundin were set as 100%. Boxplots show data of eight independent experiments and all data points are displayed. There were significant differences between the groups (ANOVA, *p* = 0.0024). * indicates significant difference from the other conditions in a *post hoc* unpaired *t*-test. (*b,c*) Visualization of lipid droplet formation by Oil Red O staining. Human astrocytes were cultured for 7 consecutive days with either 0.05% Lipofundin (*b*) or 0.005% Lipofundin (*c*). A subset of cells was cultured for 3 more days without Lipofundin in the culture media. After fixation, samples were stained with Oil Red O.

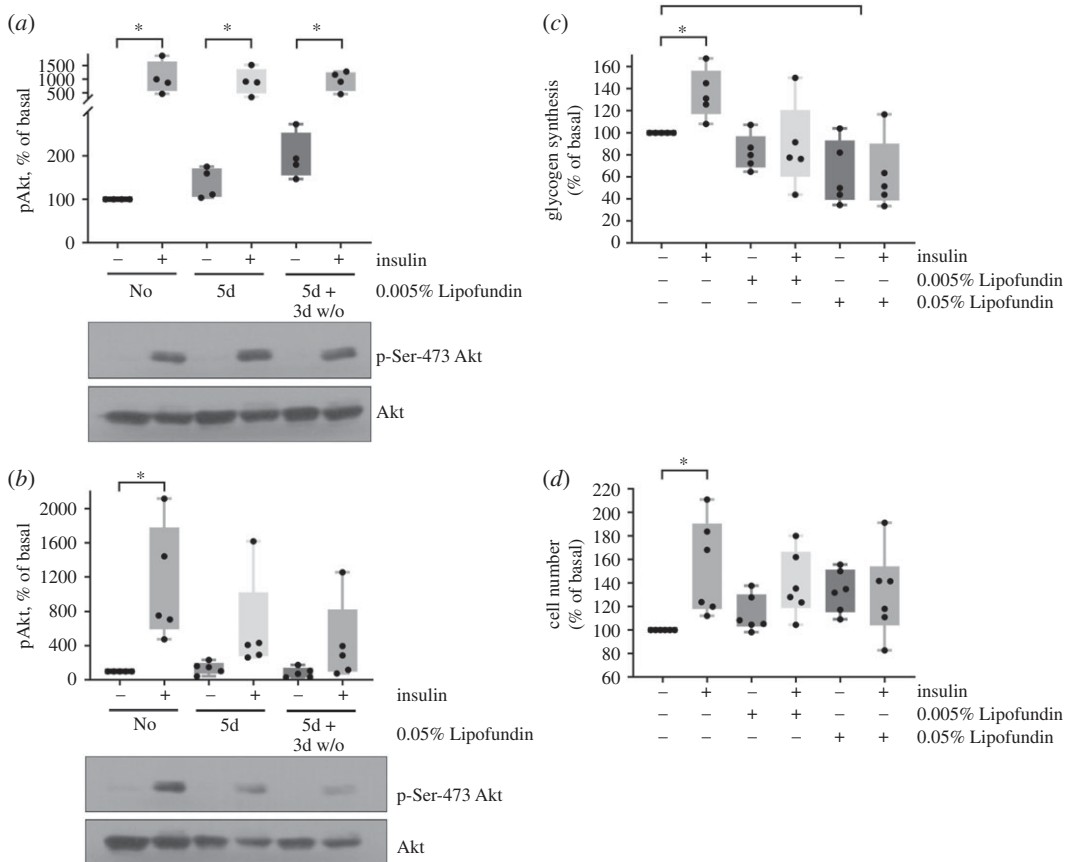

**Figure 2.** Ectopic fat storage impairs insulin-dependent activation of Akt, glycogen synthesis and proliferation. (*a,b*) Cells were cultured in medium without Lipofundin for 5 days (no) or were cultured in medium containing Lipofundin for 5 days (5d) or were kept in medium containing Lipofundin for 5 days followed by culture in Lipofundin-free medium for additional 3 days (5d + 3d w/o). In each condition, cells were left unstimulated or treated with 50 nM insulin for 30 min. In (*a*), experiments were conducted with the low concentration of Lipofundin (0.005%). There were significant differences between groups (ANOVA, $p = 0.0009$). All insulin-treated groups were significantly higher than the no Lipofundin and no insulin group. Boxplots represent datapoints from $n = 4$. In (*b*), experiments were performed with high concentration of Lipofundin (0.05%). There were significant differences between groups (ANOVA, $p = 0.0042$). Boxplots represent datapoints from $n = 5$. Akt phosphorylation of serine 473 was detected by a phospho-specific antibody, a representative western blot is shown in the lower part of *a* and *b* (phospho-specific blot and whole Akt protein). (*c*) Cells were pretreated with the indicated Lipofundin concentration for 7 days. Indicated cells were stimulated with 50 nM insulin for 3 h. Glycogen synthesis in cells untreated with Lipofundin and insulin was set as 100%. Boxplots show all datapoints; $n = 5$. There were significant differences between the groups (ANOVA, $p = 0.0018$). (*d*) Cells were pretreated with the indicated Lipofundin concentration for 7 days. Indicated cells were incubated with 50 nM insulin for another 3 days. Thereafter, cell number was estimated by WST-1 assay. Boxplots show datapoints from $n = 6$. There were significant differences between the groups (ANOVA, $p = 0.0271$). * indicates significant difference in a *post hoc* unpaired *t*-test for differences in each Lipofundin group ($p < 0.05$).

expression in astrocytes, (ii) what is the effect of continuous lipid exposure on gene expression in astrocytes, (iii) does chronic lipid exposure affect insulin-dependent gene expression, (iv) is the effect of chronic lipid exposure reversible, for both, lipid treatment alone and (v) in combination with insulin? For this, cells were exposed to various conditions as depicted in table 1. In brief, human astrocytes were cultured without any treatment as control condition (con), with 50 nM insulin for 3 h (ins), in the presence of 0.05% Lipofundin for 4 days (lipids), in the presence of 0.05% Lipofundin for 4 days followed by stimulation with 50 nM insulin for 3 h (lipids → ins), with 0.05% Lipofundin for 4 days and 3 additional days without Lipofundin (lipids → no lipids) or with 0.05% Lipofundin for 4 days and 3 additional days without Lipofundin followed by stimulation with 50 nM insulin for 3 h (lipids → no lipids → ins). We obtained the following results: (i) stimulation with insulin resulted in significant differential regulation of 485 genes compared to the unstimulated control (figure 3*b*). (ii) Chronic exposure to lipids resulted in 431 significantly changed probe sets (figure 3*c*), and

(iv) removal of lipids from the cell culture medium affected 393 genes in their expression level (data not shown). Both incubations were compared to untreated cells. Figure 3c also indicates partial reversibility of the Lipofundin treatment effect on the gene expression profile (figure 3c, left panel versus right panel). Thirty-eight reversibly regulated genes are shown in figure 3h, whereas 56 not reversible genes are depicted in figure 3g. (iii) Challenging cells for 4 consecutive days with Lipofundin before stimulation with insulin reduced the number of regulated genes to 120 when compared with Lipofundin stimulation alone (data not shown). Significantly regulated genes by insulin stimulation showed in part an inversed gene expression profile in the presence of fat (figure 3b, middle column): we observed 535 genes that were significantly regulated in the opposite direction by insulin upon additional lipid treatment (figure 3d). To assess the biological functions of these genes, we used Ingenuity Pathway Analysis software to identify enriched canonical pathways and upstream regulators. The most significantly enriched pathway was HGF signalling, others were signalling by IGF1, insulin receptor, PI3 K/AKT and AMPK (electronic supplementary material, table S1). The predicted upstream regulators comprised a number of transcription factors, including Forkhead-Box-Protein O (FOXO) 1 and 3 (electronic supplementary material, table S2). Both were predicted to be inhibited in their activity upon Lipofundin treatment, corresponding to the regulation of their 27 targets (figure 3e). Most of these targets were downregulated when cells were solely stimulated with insulin without any Lipofundin pretreatment. Among them were CCNG2 (Cyclin-G2), HBP1 (HMG-box transcription factor 1), and PIK3CA (phosphatidylinositol-4,5-bisphosphate 3-kinase catalytic subunit alpha).

(v) Omission of fatty acids from the supernatant of Lipofundin treated cells for 3 additional days before insulin stimulation resulted in 256 genes compared to lipid → no lipid alone (data not shown) and led to a partial reversal of the fatty acid effect (figure 3b left versus right column). Forty-eight genes were regulated in the same direction by insulin in both set-ups (ins versus con and lipid → no lipid → ins versus lipid → no lipid) and their expression after removal of Lipofundin from the cell culture medium for 3 days was similar to untreated cells (figure 3f).

# 4. Discussion

Astrocytes are the most abundant non-neuronal cell type in the central nervous system. Given the fact that astrocytes respond to insulin and are capable of lipid synthesis, storage and oxidation, we investigated the effect of chronic lipid oversupply on the metabolic and mitogenic effects of insulin on cultured human astrocytes. Chronic stimulation with Lipofundin resulted in ectopic lipid droplet formation in astrocytes. The cells' gene expression profile was markedly altered by insulin stimulation, but in the presence of lipids, insulin stimulation resulted in an inverse gene regulation. On a functional level, impaired activation of the insulin signalling cascade, reduced glycogen synthesis, and cell proliferation were observed in response to intracellular lipid storage.

We observed that 3 days of chronic lipid exposure led to prominent accumulation of lipid droplets in human astrocytes. Kwon et al. also stained astrocytes positive for lipid droplets after 48 h of incubation with 200 µM palmitate [29].

In contrast to their approach, we aimed to provide astrocytes with a mixture of several fatty acids instead of only one fatty acid to mimic more closely the in vivo situation. In physiology, a complex mixture of lipids is present in the plasma as well as in the cerebrospinal fluid [33]. All substances that cross the blood–brain barrier first cross the endothelial cell layer that covers blood vessels. These endothelial cells mediate transport to the attached astrocytes. Noteworthy, astrocytes are not in direct contact with the blood vessels, and concentrations of fatty acids are considerably lower in cerebrospinal fluid than in plasma. It still remains controversial if fatty acids cross the blood–brain barrier via diffusion or with specific protein-mediated transport [34]. Abdelmagid et al. measured free fatty acids in plasma of young and healthy persons with diverse ethnic background [35] and Schmid et al. measured cerebrospinal fluid fatty acids in healthy controls [33]. Concentrations of major fatty acids are summarized in table 2 and compared to the concentrations used in our experiments. Our higher Lipofundin concentration was already below physiologic plasma concentrations of fatty acids, albeit by using these low concentrations we could induce ectopic lipid storage that persisted even when Lipofundin was removed from the culture medium for several days. We therefore assume that our conditions probably resemble physiologic fatty acid concentrations that are present in the area surrounding astrocytes to exert biologically relevant effects.

(a)

| label | analysis | experimental condition | | |
|---|---|---|---|---|
| (a) | insulin effect | ins | versus | con |
| (b) | insulin effect after Lipofundin treatment | lipids → ins | versus | lipids |
| (c) | insulin effect after Lipofundin treatment and Lipofundin withdrawal | lipids → no lipids → ins | versus | lipids → no lipids |
| (d) | lipid effect | lipids | versus | con |
| (e) | lipid effect after Lipofundin withdrawal | lipids → no lipids | versus | con |

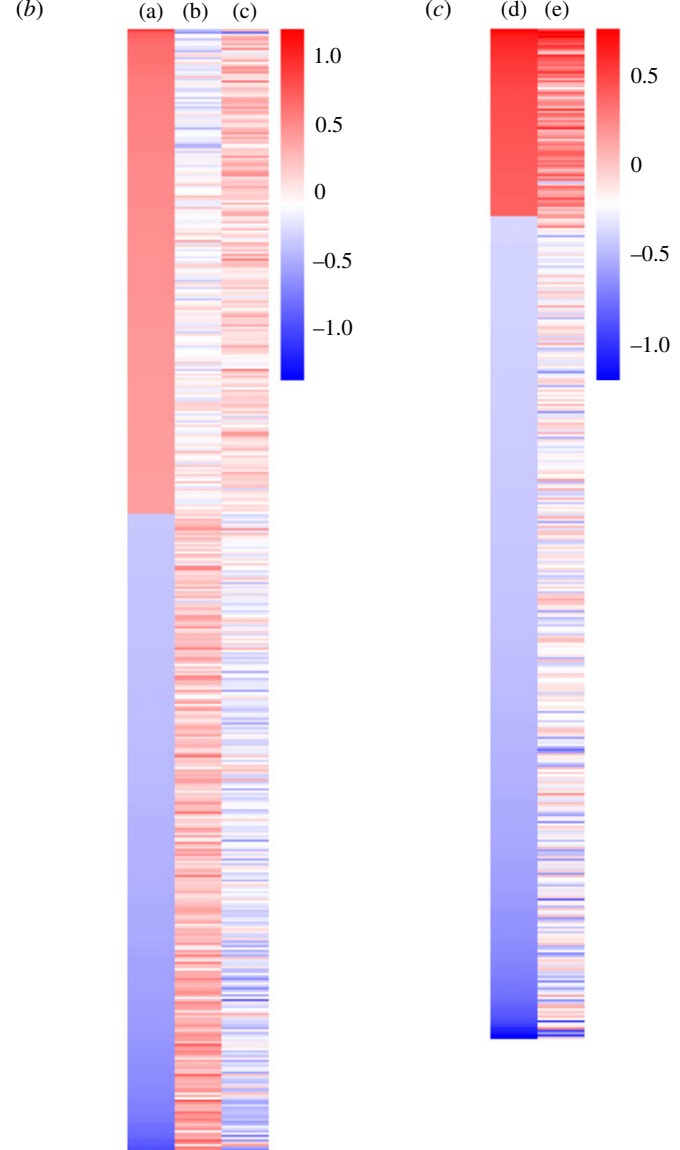

**Figure 3.** Lipofundin treatment alters gene expression profile in human astrocytes. (*a*) Depiction of analysis. (*b*) Heatmap representing the regulation of 485 genes based on comparison (a). (*c*) Heatmap showing 431 genes regulated in the comparison (d). (*d*) 535 genes with inverse regulation between ins and lipids + ins. (*e*) Predicted targets of FOXO1 and FOXO3 among this set of 535 genes shown in (*d*) Red (green) indicates up (down)-regulation by the presence of Lipofundin and bar-bell-shaped icons indicate transcription factors. (*f*) Forty-eight genes were regulated in the same direction in the two analyses ins versus con and lipids → no lipids → ins versus lipids → no lipids. (*g,h*) The comparison of the two analyses lipids versus con and lipids/no lipids versus con revealed genes, which are either not reversible (*g*, 56 genes) or reversible (*h*, 38 genes) upon lipid withdrawal. Gene expression is presented as ratios (*b,c*) or as absolute expression values (*d–g*). Scale bar indicates upregulation (red) or downregulation (blue).

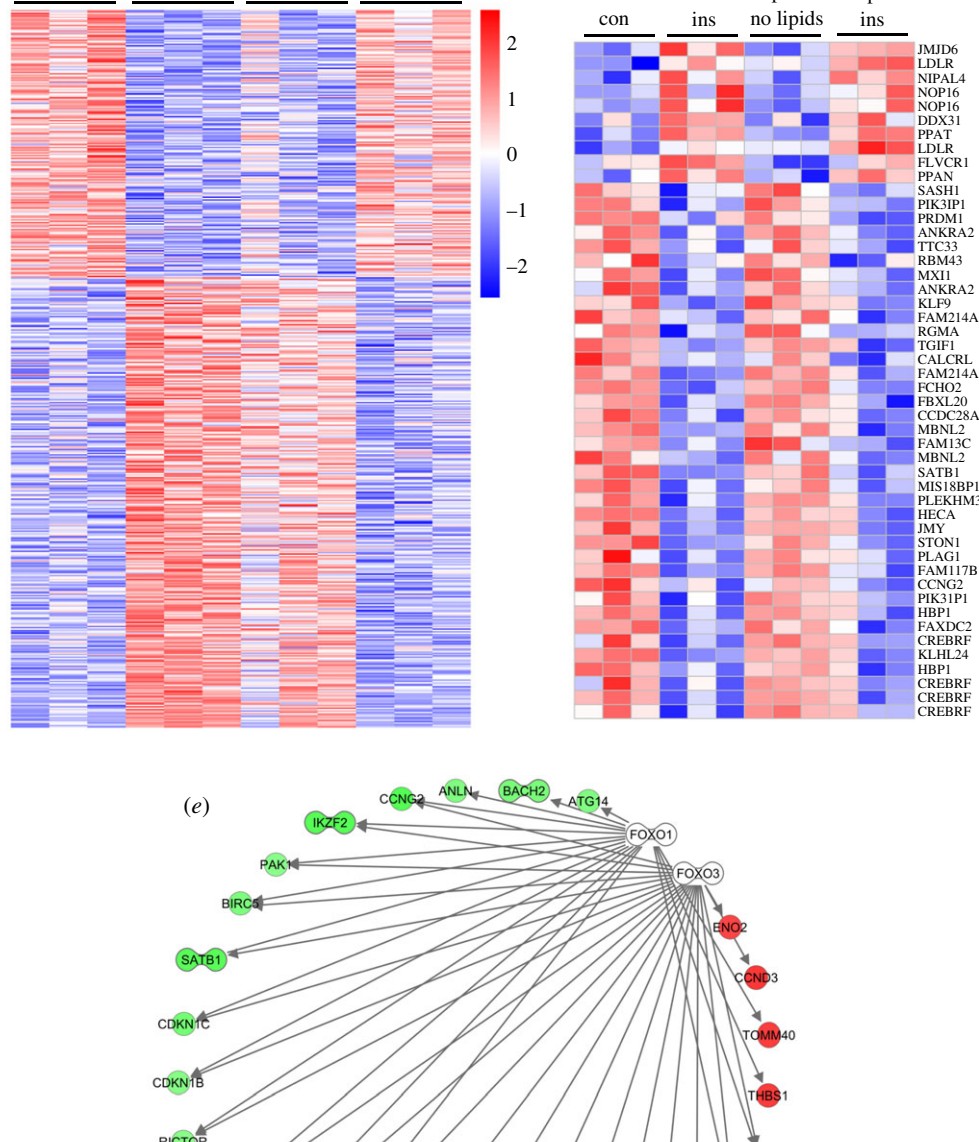

**Figure 3.** (Continued.)

We observed that ectopic lipid storage induces insulin resistance in human astrocytes similar to what was previously reported in hepatocytes and skeletal muscle cells [36,37]. Our data show significantly reduced insulin-stimulated Akt phosphorylation and subsequently impaired glycogen synthesis. Within the brain, glycogen is most abundant in astrocytes [38]. A reduced capacity to store glycogen impairs these cells' ability to acutely supply neurons with energy in the form of lactate in periods of intense neural activity or hypoglycaemia [17]. Studies from Suzuki *et al.* and Newman *et al.* revealed that astrocytic-derived lactate from glycogenolysis is crucial for long-term memory formation and thus for higher brain function [39,40]. Furthermore, Duran and colleagues showed that genetic ablation of glycogen synthase in the central nervous system of mice results in impaired hippocampal long-term

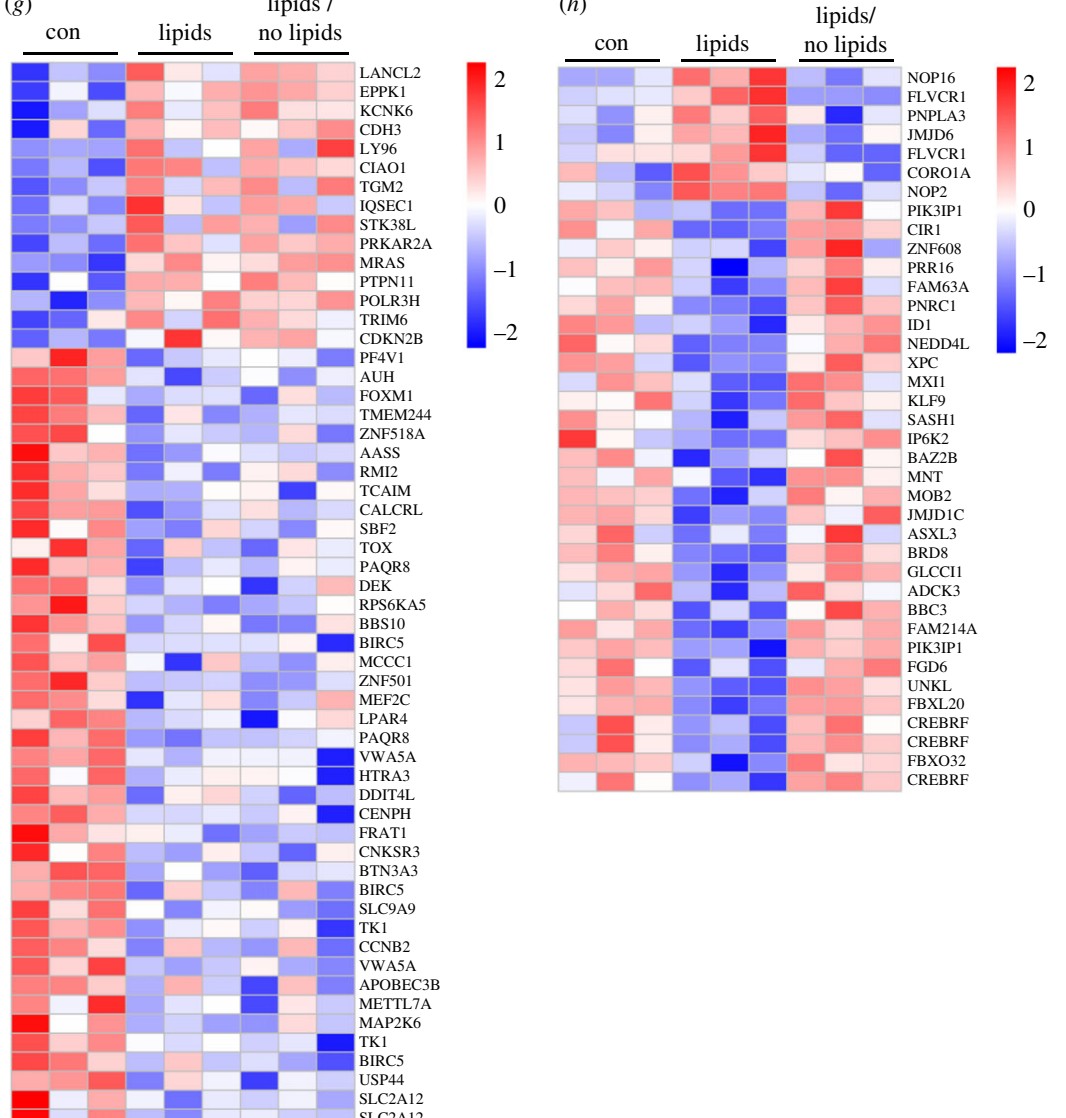

**Figure 3.** (Continued.)

memory formation and associative learning [41]. Interestingly, this specific aspect of memory is modulated by insulin in humans and might therefore be affected by insulin resistance due to ectopic lipid accumulation in astrocytes. Animal studies further implicate that hippocampal insulin resistance could be involved in the cognitive impairment that is observed in experimental models of diabetes [42]. Of note, our data indicate that not only insulin-dependent glycogen synthesis is diminished in astrocytes that store excessive fatty acids, but also basal glycogen synthesis was significantly reduced in these glia cells. As the astrocytic glycogen pool is highly dynamic, the observed fat-induced impairment in astrocytic glycogen metabolism might contribute to the generally disturbed brain metabolism in obesity.

The observed insulin resistance in our cultured astrocytes is probably a result of effects mediated by intracellular ectopic fat storage. Insulin resistance induced by lipid oversupply can be mediated by several pathways. Saturated fatty acids signal via toll-like receptors (TLR) 2 and 4. Studies from our department in primary astrocytes from TLR2/4 knock-out mice have shown increased phosphorylation of p-Akt and p-GSK after insulin stimulation and knockout animals were protected from brain insulin resistance on high-fat diet [43]. In addition, ectopic fat storage leads to the formation and accumulation of metabolic lipid intermediates like ceramides and DAGs (diacylglycerol) that inhibit insulin signalling pathways [44].

When we analysed insulin-induced proliferation in astrocytes, we observed an increase in cell number in cells incubated with the high Lipofundin (0.05%) concentration. Our finding is in line with

**Table 2.** Concentrations of representative fatty acids in serum [35], cerebrospinal fluid [33] and the tissue culture experiments in this study.

| | serum | cerebrospinal fluid | 0.05% Lipofundin | 0.005% Lipofundin |
|---|---|---|---|---|
| palmitic acid | 0.3–4.1 mM | 0.642–5.13 µM | 16–33.5 µM | 1.6–33.5 µM |
| linolenic acid | 0.2–5 mM | 2.7–13.9 µM | 85.5–103.5 µM | 8.55–10.35 µM |
| α-linoleic acid | 12–187 µM | 0.018–0.61 nM | 9–19.8 µM | 0.9–1.98 µM |

the literature, where high-fat diet in rodents induces cell proliferation of astrocytes [45–48]. However, insulin was not able to further increase proliferation when cells were cultured in the presence of Lipofundin. This finding might be due to a ceiling effect in our cell culture system where insulin could not increase the astrocytic cell number any further. In general, astrocytes respond to external stimuli mainly by induction of reactive gliosis and to a lesser extent by proliferation [49]. In the physiological situation, astrocytic response to high-fat conditions seems to depend on the location of the astrocyte within the brain. Buckman *et al.* for example described regional differences in GFAP immunoreactivity in hypothalamus of high-fat fed mice [47]). In addition, the cell culture conditions we used did not introduce an inflammatory environment for the astrocytes that was described earlier to stimulate cell proliferation [45–47]. It is also conceivable that other factors released from cells adjacent to astrocytes *in vivo* are necessary to trigger cell proliferation in astrocytes.

Several questions regarding the effect of insulin and lipids on astrocytic gene expression patterns were addressed. Different treatment of human astrocytes with Lipofundin and insulin resulted in differential regulation of genes. The presence of Lipofundin altered insulin-mediated gene expression and the analysis of associated canonical pathways resulted in several enriched signalling pathways. Among them were insulin receptor signalling and PI3 K/AKT signalling pathways, which corresponds to our experimental data. Of note, several genes showed a reversible expression pattern upon withdrawal of Lipofundin for 3 days, whereas some did not. The analysis for predicted upstream regulators of the significantly differently expressed genes revealed, among others, the transcription factors FOXO1 and FOXO3. FOXOs are well-known transcription factors for insulin-dependent metabolic regulation. Genes regulated by FOXO1 and FOXO3 in astrocytes were decreased in their expression upon insulin treatment for 3 h, thus indicating a potential negative feedback mechanism to turn off the insulin signal at this time point. Interestingly, the same genes showed an upregulation when cells were stimulated with Lipofundin for 4 consecutive days before insulin stimulation was conducted. Thus, chronic lipid exposure impairs insulin-stimulated gene expression, indicating insulin resistance also at the level of FOXO-dependent transcription.

Among the limitations of our current experiments are the different timeframes of our cells in culture that might have influenced cell number and could have an impact on our results, especially for the condition with subsequent Lipofundin withdrawal. Another limitation is that we were not able to collect more information on the activated insulin signalling cascade upstream of Akt, for example, insulin receptor phosphorylation, as it was not reliably possible to detect this in western blots, presumably due to the limited protein amount available.

A sedentary lifestyle accompanied by chronic nutrient oversupply results in ectopic fat storage in peripheral tissues like skeletal muscle and the liver. Ectopic fat storage impairs metabolic health as it is a major driver of the development of insulin resistance. Research over the last years has shown that insulin resistance is not limited to the periphery, but also present in the brain [4,5,50]. Brain insulin resistance has negative effects on peripheral insulin sensitivity as this condition impairs brain-derived signals that improve metabolism in peripheral tissues [9,51]. However, the underlying mechanisms of brain insulin resistance in obesity are still not fully understood. It is well accepted that many neural processes require the participation of astrocytes. Astrocytes function as mediators in the communication between the periphery and the neurons and are crucial for neuronal survival [16]. Studies in various mouse models have established the biological relevance for storage of fatty acids in lipid droplets in the brain [52–54]. However, the role of astrocytes in lipid storage and homeostasis of the brain needs further clarification to understand consequences of potentially pathologic changes in astrocytic lipid metabolism. We investigated how chronic lipid supply to human astrocytes affects insulin-dependent mechanisms. Our data indicate that ectopic fat storage leads to insulin resistance, impaired glycogen storage and reduced proliferation in cultured human astrocytes. This could

contribute to the insulin resistance of the entire organ that was detected in humans. Presumably, compromised insulin signalling in the brain results in negative effects on neuronal plasticity, survival and memory [55]. An important question is, if astrocytic insulin resistance is reversible. Substantial weight loss can reverse brain insulin resistance in rats [56]. However, the involved cell types and the transferability to the human situation are still unclear. Our current analyses of the gene expression pattern in astrocytes indicate that astrocytic insulin resistance might at least in parts be reversible. Given the importance of astrocytes within the central nervous system [23], overcoming astrocytic insulin resistance by reducing ectopic lipid load might represent one promising upcoming treatment target for insulin resistance of the brain in obesity, diabetes and neurodegeneration.

Data accessibility. All relevant data are included in the manuscript. Additional data can be found in the electronic supplementary material.

Authors' contributions. M.H. researched and analysed data and drafted the manuscript. S.S.E. analysed data and drafted the manuscript. J.S., A.B., and N.H. researched data and contributed to discussion. M.I. analysed gene expression data and contributed to discussion. J.B., M.H.A., H.-U.H. and A.F. contributed to discussion. H.S. designed experiments and contributed to discussion. All authors approved the final version of the manuscript prior to submission.

Competing interests. There are no competing interests.

Funding. This study was supported in part by a grant from the German Federal Ministry of Education and Research (BMBF) to the German Centre for Diabetes Research (DZD e.V.,: 01GI0925) and the Helmholtz Alliance 'Aging and Metabolic Programming, AMPro' (J.B.).

Acknowledgements. We gratefully acknowledge the excellent technical assistance of Leonie Nono and Carina Hermann from the IDM in Tübingen, Germany. We also thank Dr Felicia Gerst (IDM). We acknowledge support by the Deutsche Forschungsgemeinschaft and the Open Access Publishing Fund of the University of Tübingen, Germany.

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
