## [Reviewer comments · Royal Society Open Science]

Review History

RSOS-200701.R0 (Original submission)

Review form: Reviewer 1

Is the manuscript scientifically sound in its present form?

Yes

Are the interpretations and conclusions justified by the results?

Yes

Is the language acceptable?

No

Do you have any ethical concerns with this paper?

No

Have you any concerns about statistical analyses in this paper?

Yes

Recommendation?

Major revision is needed (please make suggestions in comments)

Comments to the Author(s)

In this manuscript, Heni et al. provide evidence that lipid accumulation in cultured human astrocytes leads to aberrant gene expression and attenuated responsiveness to insulin. Overall, the question authors attempted to address is timely and important, and they obtained an interesting set of results. Nonetheless, I found the manuscript confusing and suggest some improvements below:

- 1) In terms of data presentation, Figures 1 and 2 could be combined. Most importantly, given that insulin resistance is usually assessed through phosphorylation of target proteins, the Western blots demonstrating that insulin-induced Akt phosphorylation is attenuated by lipid accumulation (Fig 4) should appear before gene expression data (Fig 3).
- 2) The methodology is not clear enough regarding treatments. Some conditions appear to have remained in culture for longer periods. It may be inappropriate to compare a condition that was treated for only 3 hours with other that remained in culture for ten more days. How can authors account for cellular aging or confluence with such a difference in time of collection? Indeed, most treatments seemed to slightly increase cell density (Fig 4). If that was not the case, authors should rephrase their experimental design to explain what was done more clearly.
- 3) It is impressive that authors found such big effects of lipids on basal and insulin-induced gene expression. Here, I would suggest modifying the nomenclature used in the text for each condition. As is, this section does not clearly inform. The authors should also consider a scheme to depict the experimental conditions.
- 4) Still, this manuscript narrows too much on the number of DEGs instead of their roles/pathways. I strongly suggest that authors perform a gene ontology/KEGG analysis with their present data. This would bring relevant additional information for the current findings. Then, authors will be able to discuss how the interplay between fatty acids and insulin signalling on the modulation of intracellular pathways relevant for astrocyte physiology.
- 5) Finally, the discussion could be improved by adding some recent related literature (Ferris et al., PNAS, 2017; Melo et al., Cell Reports, 2020; Kim et al., Neurobiol Dis, 2020). Findings on brain insulin resistance and metabolic dysfunction could also be discussed (Biessels & Reagan, Nat Rev Neurosci, 2015; Clarke et al., J Alz Dis, 2018).
- 6) Statistics: Given there are multiple conditions in this experimental design, it is important to ensure proper statistical analyses were done. The authors should indicate whether one-way or two-way ANOVA was conducted, whether they are two-sided, which post-hoc tests were used, etc.

Review form: Reviewer 2

Is the manuscript scientifically sound in its present form?

No

Are the interpretations and conclusions justified by the results?

No

Is the language acceptable?

Yes

Do you have any ethical concerns with this paper?

No

Have you any concerns about statistical analyses in this paper?

Yes

Recommendation?

Major revision is needed (please make suggestions in comments)

Comments to the Author(s)

In this study, the authors aimed to investigate the effects of ectopic lipid accumulation in human astrocytes in the insulin signaling in vitro. To do so, Heni and colleagues cultivated human astrocytes (Lonza) in the presence of a lipid emulsion (Lipofundin®) for seven days and executed cell culture, lipid stain, gene expression, and western blotting experiments. Chronic exposure to lipid emulsion increased histological visualization of ectopic fat storage in astrocytes after three days, which remains until the seventh day of treatment and for another three days without Lipofundin®.

To understand how lipid exposure affected gene expression in astrocytes, the transcriptome profiling was analyzed in response to insulin stimuli (3 hours) in the presence of Lipofundin® or after Lipofundin® was withdrawn from the culture medium. Results showed that gene expression was affected diversely in response to lipids and/or insulin, being reversible or not with lipid removal. In this line, Heni and colleagues highlighted the inverse expression profile of some genes in response to insulin stimuli in the presence or absence of lipid emulsion. Moreover, some genes were regulated by insulin regardless of the presence of lipids. Among such genes, several were regulated by the transcription factors FOXO 1 and 3, a known target of the insulin signaling pathway. Thus, to evaluate astrocytic insulin signaling, the authors investigated insulin-induced phosphorylation of Akt, glycogen synthesis, and cell proliferation after lipid exposure (four days) or three days after removal of Lipofundin® from the cell culture medium. The authors suggested insulin signaling impairment as a result of ectopic lipid accumulation.

Although the effect of excessive lipid exposure and the formation of astrocytic lipid droplets is not a new concept, the analysis of the expression profile in response to the presence or absence of the lipid emulsion suggested possible reversibility of lipid toxicity in the astrocytic insulin signaling dysfunction. The manuscript focused on correlating lipid accumulation with insulin signaling dysfunction, but some points still need to be reinforced:

- 1) The human astrocyte culture needs to be better characterized. How is the expression of classical in vitro cultured astrocyte markers (ALDH1L1, EAAT1, EAAT2, GFAP)? For example, what is the percentage of GFAP positive cells? What was the number of cells used in the experiments? Doesn't the use of human cells, even if commercially obtained, require any type of approval for experimentation?
- 2) It was not clear whether the treatment with the lipid emulsion was done. Lipofundin® was used only once or repeated with each change of cell medium?
- 3) The cell viability test using lactate dehydrogenase activity was analyzed after five days of treatment. How is the viability of the human astrocyte culture after seven days of Lipofundin® treatment and after Lipofundin® was withdrawn from the cell culture medium?

Although lactate dehydrogenase (LDH) released into the medium was measured as a marker of cell death, it is a cell viability measure. If authors would like to express cell death, maybe consider the use of another complementary analysis, for example live/dead assay.

How was the LDH release calculated about basal conditions? One suggestion is to express the data as dot-plot and insert the error bar in the control (basal) condition to represent the variability between different cultures. Moreover, is not clear how statistics analysis was performed. Which post-hoc test was used, and which comparisons were made?

4) How was the Lipofundin® withdrawal protocol? Was there any washing before change cell culture media?

5) In figure 2, more information is needed on: microscope and objective used, scale bar, how many cultures were used and how many images for each culture was analyzed. Is the same cell culture seen over time or different cell cultures in each time point?

In the "no more Lipofundin" images, check the number of days on the panels.

6) In figure 3, the heat-maps have a nice big picture of the effects observed in different experimental conditions. Perhaps, a graphical (maybe a volcano plot?) showing the variation of the genes related to insulin signaling (regulated by FOXO) would highlight the focus of the manuscript discussion and statistical analysis used to include genes.

7) In figure 4, how did the levels of Akt phosphorylation be normalized? How are the total Akt levels in the different conditions compared to the baseline? Was there a loading control for these blottings? Which post-hoc test was done and how were the comparisons made between the different conditions?

One suggestion is to express the data as dot-plot and insert the error bar in the control (basal) condition to represent the variability between different cultures.

8) Considering the interest in the correlation between the ectopic lipid accumulation and insulin signaling impairment, it would be interesting to assess the total and phosphorylation levels of the insulin receptor.

9) As a general suggestion, more technical information could be added to the methodology section, which would facilitate the understanding and reproducibility of the experiments. In addition, the figure legends could be more objective and bring more information about the panel data.

Decision letter (RSOS-200701.R0)

Dear Dr Heni,

The editors assigned to your paper ("Ectopic fat accumulation in human astrocytes impairs insulin action") have now received comments from reviewers. We would like you to revise your paper in accordance with the referee and Associate Editor suggestions which can be found below (not including confidential reports to the Editor). Please note this decision does not guarantee eventual acceptance.

Please submit a copy of your revised paper before 21-Jun-2020. Please note that the revision deadline will expire at 00.00am on this date. If we do not hear from you within this time then it will be assumed that the paper has been withdrawn. In exceptional circumstances, extensions

may be possible if agreed with the Editorial Office in advance. We do not allow multiple rounds of revision so we urge you to make every effort to fully address all of the comments at this stage. If deemed necessary by the Editors, your manuscript will be sent back to one or more of the original reviewers for assessment. If the original reviewers are not available, we may invite new reviewers.

- Data accessibility

If you wish to submit your supporting data or code to Dryad (<http://datadryad.org/>), or modify your current submission to dryad, please use the following link:
<http://datadryad.org/submit?journalID=RSOS&manu=RSOS-200701>

- Competing interests

- Authors' contributions

- Acknowledgements

- Funding statement

Kind regards,
Lianne Parkhouse
Editorial Coordinator
Royal Society Open Science
openscience@royalsociety.org

on behalf of Dr Robson da Costa (Associate Editor) and Catrin Pritchard (Subject Editor)
openscience@royalsociety.org

Associate Editor's comments (Dr Robson da Costa):

The MS has been assessed by two expert referees in the field. As you can see in the referee reports, the MS needs major revision, particularly in the description of the methods, data presentation and statistical analysis. Concerns about the ethical considerations exist, please make it clear. Addressing these issues will make this paper more impactful.

Reviewers' Comments to Author:

Reviewer: 1

Comments to the Author(s)

In this manuscript, Heni et al. provide evidence that lipid accumulation in cultured human astrocytes leads to aberrant gene expression and attenuated responsiveness to insulin. Overall, the question authors attempted to address is timely and important, and they obtained an interesting set of results. Nonetheless, I found the manuscript confusing and suggest some improvements below:

1) In terms of data presentation, Figures 1 and 2 could be combined. Most importantly, given that insulin resistance is usually assessed through phosphorylation of target proteins, the Western blots demonstrating that insulin-induced Akt phosphorylation is attenuated by lipid accumulation (Fig 4) should appear before gene expression data (Fig 3).

2) The methodology is not clear enough regarding treatments. Some conditions appear to have remained in culture for longer periods. It may be inappropriate to compare a condition that was treated for only 3 hours with other that remained in culture for ten more days. How can authors account for cellular aging or confluence with such a difference in time of collection? Indeed, most treatments seemed to slightly increase cell density (Fig 4). If that was not the case, authors should rephrase their experimental design to explain what was done more clearly.

3) It is impressive that authors found such big effects of lipids on basal and insulin-induced gene expression. Here, I would suggest modifying the nomenclature used in the text for each condition. As is, this section does not clearly inform. The authors should also consider a scheme to depict the experimental conditions.

4) Still, this manuscript narrows too much on the number of DEGs instead of their roles/pathways. I strongly suggest that authors perform a gene ontology/KEGG analysis with their present data. This would bring relevant additional information for the current findings. Then, authors will be able to discuss how the interplay between fatty acids and insulin signalling on the modulation of intracellular pathways relevant for astrocyte physiology.

5) Finally, the discussion could be improved by adding some recent related literature (Ferris et al., PNAS, 2017; Melo et al., Cell Reports, 2020; Kim et al., Neurobiol Dis, 2020). Findings on brain insulin resistance and metabolic dysfunction could also be discussed (Biessels & Reagan, Nat Rev Neurosci, 2015; Clarke et al., J Alz Dis, 2018).

6) Statistics: Given there are multiple conditions in this experimental design, it is important to ensure proper statistical analyses were done. The authors should indicate whether one-way or two-way ANOVA was conducted, whether they are two-sided, which post-hoc tests were used, etc.

Reviewer: 2

Comments to the Author(s)

In this study, the authors aimed to investigate the effects of ectopic lipid accumulation in human astrocytes in the insulin signaling in vitro. To do so, Heni and colleagues cultivated human astrocytes (Lonza) in the presence of a lipid emulsion (Lipofundin®) for seven days and executed cell culture, lipid stain, gene expression, and western blotting experiments. Chronic exposure to lipid emulsion increased histological visualization of ectopic fat storage in astrocytes after three days, which remains until the seventh day of treatment and for another three days without Lipofundin®.

To understand how lipid exposure affected gene expression in astrocytes, the transcriptome profiling was analyzed in response to insulin stimuli (3 hours) in the presence of Lipofundin® or after Lipofundin® was withdrawn from the culture medium. Results showed that gene expression was affected diversely in response to lipids and/or insulin, being reversible or not with lipid removal. In this line, Heni and colleagues highlighted the inverse expression profile of some genes in response to insulin stimuli in the presence or absence of lipid emulsion. Moreover, some genes were regulated by insulin regardless of the presence of lipids. Among such genes, several were regulated by the transcription factors FOXO 1 and 3, a known target of the insulin signaling pathway. Thus, to evaluate astrocytic insulin signaling, the authors investigated insulin-induced phosphorylation of Akt, glycogen synthesis, and cell proliferation after lipid exposure (four days) or three days after removal of Lipofundin® from the cell culture medium. The authors suggested insulin signaling impairment as a result of ectopic lipid accumulation.

Although the effect of excessive lipid exposure and the formation of astrocytic lipid droplets is not a new concept, the analysis of the expression profile in response to the presence or absence of the lipid emulsion suggested possible reversibility of lipid toxicity in the astrocytic insulin signaling dysfunction. The manuscript focused on correlating lipid accumulation with insulin signaling dysfunction, but some points still need to be reinforced:

1) The human astrocyte culture needs to be better characterized. How is the expression of classical in vitro cultured astrocyte markers (ALDH1L1, EAAT1, EAAT2, GFAP)? For example, what is the percentage of GFAP positive cells? What was the number of cells used in the experiments? Doesn't the use of human cells, even if commercially obtained, require any type of approval for experimentation?

2) It was not clear whether the treatment with the lipid emulsion was done. Lipofundin® was used only once or repeated with each change of cell medium?

3) The cell viability test using lactate dehydrogenase activity was analyzed after five days of treatment. How is the viability of the human astrocyte culture after seven days of Lipofundin® treatment and after Lipofundin® was withdrawn from the cell culture medium?

Although lactate dehydrogenase (LDH) released into the medium was measured as a marker of cell death, it is a cell viability measure. If authors would like to express cell death, maybe consider the use of another complementary analysis, for example live/dead assay.

How was the LDH release calculated about basal conditions? One suggestion is to express the data as dot-plot and insert the error bar in the control (basal) condition to represent the variability between different cultures. Moreover, is not clear how statistics analysis was performed. Which post-hoc test was used, and which comparisons were made?

4) How was the Lipofundin® withdrawal protocol? Was there any washing before change cell culture media?

5) In figure 2, more information is needed on: microscope and objective used, scale bar, how many cultures were used and how many images for each culture was analyzed. Is the same cell culture seen over time or different cell cultures in each time point?

In the "no more Lipofundin" images, check the number of days on the panels.

6) In figure 3, the heat-maps have a nice big picture of the effects observed in different experimental conditions. Perhaps, a graphical (maybe a volcano plot?) showing the variation of the genes related to insulin signaling (regulated by FOXO) would highlight the focus of the manuscript discussion and statistical analysis used to include genes.

7) In figure 4, how did the levels of Akt phosphorylation be normalized? How are the total Akt levels in the different conditions compared to the baseline? Was there a loading control for these blottings? Which post-hoc test was done and how were the comparisons made between the different conditions?

One suggestion is to express the data as dot-plot and insert the error bar in the control (basal) condition to represent the variability between different cultures.

8) Considering the interest in the correlation between the ectopic lipid accumulation and insulin signaling impairment, it would be interesting to assess the total and phosphorylation levels of the insulin receptor.

9) As a general suggestion, more technical information could be added to the methodology section, which would facilitate the understanding and reproducibility of the experiments. In addition, the figure legends could be more objective and bring more information about the panel data.

Author's Response to Decision Letter for (RSOS-200701.R0)

See Appendix A.

RSOS-200701.R1 (Revision)

Review form: Reviewer 1

Is the manuscript scientifically sound in its present form?

Yes

Are the interpretations and conclusions justified by the results?

Yes

Is the language acceptable?

Yes

Do you have any ethical concerns with this paper?

No

Have you any concerns about statistical analyses in this paper?

No

Recommendation?

Accept as is

Comments to the Author(s)

The authors adequately addressed all my comments.

Review form: Reviewer 2

Is the manuscript scientifically sound in its present form?

Yes

Are the interpretations and conclusions justified by the results?

Yes

Is the language acceptable?

Yes

Do you have any ethical concerns with this paper?

No

Have you any concerns about statistical analyses in this paper?

Yes

Recommendation?

Accept with minor revision (please list in comments)

Comments to the Author(s)

In this manuscript, Heni et al. showed that chronic exposure to lipids induced ectopic fat storage in cultured human astrocytes that result in a reduction in insulin signaling response and altered gene expression profile in these cells. The subject addressed by authors is important because understanding how changes in metabolism can affect astrocyte function has been shown to be relevant in several areas and diseases, as obesity, diabetes, and neurodegenerative disorders. The authors made several improvements regarding mainly the methodological and statistical

description of the results, as well as the organization of the manuscript, highlighting the limitations related to the differences in treatment times and analyses performed.

In this line, I still suggest some other improvements:

1. It would be interesting to insert a sentence related to quality control and ethical approval for the use of commercially obtained cells for the analyzes described in this study in the "Cell Culture" section (Materials and Methods).
2. Data now is present as boxplots with individual data points. The graphic representation of the data as a bar graph or boxplot can be chosen by the authors. The previous suggestion was just to insert the data as individual points to allow the visualization of the variation between the different cultures analyzed. Depending on the choice of the type of graph, it is necessary to revise the legends and adapt the description.
3. Considering that the images shown in figure 2B-C are representative, it is relevant to describe the number of cultures in which the ectopic accumulation of lipids was observed or even if it was a single observation. I also suggest inserting the type of microscope and magnification used to image acquisition in the materials and methods section or figure legend.
4. The authors improved the description of the statistical analyzes used in this manuscript. However, in figures 1 and 2 is not clear why each panel/figure used a different ANOVA post-hoc test. Moreover, the lines used to indicate statistical comparison between groups in the graphs are a little confused, mainly when more than one type of statistical analysis was used between different experimental conditions (for example, Figure 2). I suggest inserting a sentence explaining the choice of different post-hoc tests and the adjustment of the lines that indicate the comparisons in the graphs.

Decision letter (RSOS-200701.R1)

Dear Miss Heni:

On behalf of the Editors, I am pleased to inform you that your Manuscript RSOS-200701.R1 entitled "Ectopic fat accumulation in human astrocytes impairs insulin action" has been accepted for publication in Royal Society Open Science subject to minor revision in accordance with the referee suggestions. Please find the referees' comments at the end of this email.

The reviewers and Subject Editor have recommended publication, but also suggest some minor revisions to your manuscript. Therefore, I invite you to respond to the comments and revise your manuscript.

- Ethics statement

- Data accessibility

If you wish to submit your supporting data or code to Dryad (<http://datadryad.org/>), or modify your current submission to dryad, please use the following link:
<http://datadryad.org/submit?journalID=RSOS&manu=RSOS-200701.R1>

- **Competing interests**

- **Authors' contributions**

- **Acknowledgements**

- **Funding statement**

Because the schedule for publication is very tight, it is a condition of publication that you submit the revised version of your manuscript before 06-Aug-2020. Please note that the revision deadline will expire at 00.00am on this date. If you do not think you will be able to meet this date please let me know immediately.

To revise your manuscript, log into <https://mc.manuscriptcentral.com/rsos> and enter your Author Centre, where you will find your manuscript title listed under "Manuscripts with Decisions". Under "Actions," click on "Create a Revision." You will be unable to make your

revisions on the originally submitted version of the manuscript. Instead, revise your manuscript and upload a new version through your Author Centre.

on behalf of Dr Robson da Costa (Associate Editor) and Catrin Pritchard (Subject Editor)
openscience@royalsociety.org

Associate Editor Comments to Author (Dr Robson da Costa):
I would like to suggest the comments/suggestions made by the 2nd reviewer are fully addressed. Addressing these issues will make this paper more impactful.

Reviewer comments to Author:

Reviewer: 1

Comments to the Author(s)

The authors adequately addressed all my comments.

Reviewer: 2

Comments to the Author(s)

In this manuscript, Heni et al. showed that chronic exposure to lipids induced ectopic fat storage in cultured human astrocytes that result in a reduction in insulin signaling response and altered gene expression profile in these cells. The subject addressed by authors is important because understanding how changes in metabolism can affect astrocyte function has been shown to be relevant in several areas and diseases, as obesity, diabetes, and neurodegenerative disorders. The authors made several improvements regarding mainly the methodological and statistical description of the results, as well as the organization of the manuscript, highlighting the limitations related to the differences in treatment times and analyses performed.

In this line, I still suggest some other improvements:

1. It would be interesting to insert a sentence related to quality control and ethical approval for the use of commercially obtained cells for the analyzes described in this study in the "Cell Culture" section (Materials and Methods).
2. Data now is present as boxplots with individual data points. The graphic representation of the data as a bar graph or boxplot can be chosen by the authors. The previous suggestion was just to insert the data as individual points to allow the visualization of the variation between the different cultures analyzed. Depending on the choice of the type of graph, it is necessary to revise the legends and adapt the description.
3. Considering that the images shown in figure 2B-C are representative, it is relevant to describe the number of cultures in which the ectopic accumulation of lipids was observed or even if it was a single observation. I also suggest inserting the type of microscope and magnification used to image acquisition in the materials and methods section or figure legend.
4. The authors improved the description of the statistical analyzes used in this manuscript. However, in figures 1 and 2 is not clear why each panel/figure used a different ANOVA post-hoc test. Moreover, the lines used to indicate statistical comparison between groups in the graphs are a little confused, mainly when more than one type of statistical analysis was used between different experimental conditions (for example, Figure 2). I suggest inserting a sentence explaining the choice of different post-hoc tests and the adjustment of the lines that indicate the comparisons in the graphs.

Author's Response to Decision Letter for (RSOS-200701.R1)

See Appendix B.

Decision letter (RSOS-200701.R2)

Dear Dr Heni,

It is a pleasure to accept your manuscript entitled "Ectopic fat accumulation in human astrocytes impairs insulin action" in its current form for publication in Royal Society Open Science.

Kind regards,
Lianne Parkhouse
Editorial Coordinator
Royal Society Open Science
openscience@royalsociety.org

on behalf of Dr Robson da Costa (Associate Editor) and Professor Catrin Pritchard (Subject Editor)
openscience@royalsociety.org

Appendix A

Response to Reviewers

We thank you for giving us the opportunity to revise and improve our manuscript.

We thank the reviewers for the encouraging comments. We feel that their helpful suggestions substantially improved the manuscript. Please find below a point-to-point response. Changes in the manuscript are marked as underlined.

Reviewer: 1

Comments to the Author(s)

In this manuscript, Heni et al. provide evidence that lipid accumulation in cultured human astrocytes leads to aberrant gene expression and attenuated responsiveness to insulin. Overall, the question authors attempted to address is timely and important, and they obtained an interesting set of results. Nonetheless, I found the manuscript confusing and suggest some improvements below:

1) In terms of data presentation, Figures 1 and 2 could be combined. Most importantly, given that insulin resistance is usually assessed through phosphorylation of target proteins, the Western blots demonstrating that insulin-induced Akt phosphorylation is attenuated by lipid accumulation (Fig 4) should appear before gene expression data (Fig 3).

As suggested by the reviewer, we now merged the former Fig 1 with former Fig 2 together to a new Fig 1 A-C and updated the numbering and Figure legends accordingly. In addition, we switched places of the figures regarding change in gene expression with the figures showing Akt phosphorylation, glycogen synthesis and proliferation and also adapted the sections in the methods and the discussion.

2) The methodology is not clear enough regarding treatments. Some conditions appear to have remained in culture for longer periods. It may be inappropriate to compare a condition that was treated for only 3 hours with other that remained in culture for ten more days. How can authors account for cellular aging or confluence with such a difference in time of collection? Indeed, most treatments seemed to slightly increase cell density (Fig 4). If that was not the case, authors should rephrase their experimental design to explain what was done more clearly.

Thank you for this comment. To better clarify the treatment conditions, we now added Table 2 to the manuscript. We agree that we cannot rule out time effects for some conditions that might have additionally influenced cell numbers. This is now discussed in the limitations part of the manuscript.

3) It is impressive that authors found such big effects of lipids on basal and insulin-induced gene expression. Here, I would suggest modifying the nomenclature used in the text for each condition. As is, this section does not clearly inform. The authors should also consider a scheme to depict the experimental conditions.

We agree that many experimental conditions make nomenclature fairly complicated and we thank the reviewer for the good point to improve this. To this end, we added Table 2 where we depict the experimental conditions and we also modified the abbreviated conditions in the main text. In addition, the performed analyses are shown in a tabular view in Fig 3A for a better understanding of the data.

4) Still, this manuscript narrows too much on the number of DEGs instead of their roles/pathways. I strongly suggest that authors perform a gene ontology/KEGG analysis with their present data. This would bring relevant additional information for the current findings. Then, authors will be able to discuss how the interplay between fatty acids and insulin signaling on the modulation of intracellular pathways relevant for astrocyte physiology.

We thank the reviewer for this good point. In a new analysis we compared the ratios of genes that were altered in their expression by insulin treatment with ratios of genes that were altered in their expression by treatment of cells for 4 days with Lipofundin and subsequent stimulation of cells for 30 minutes with insulin. By doing so, we looked at the insulin effect and how lipid treatment alters this insulin effect. This analysis is visualized in Fig 3D and FOXO1 and FOXO3 with subsequently regulated targets as example for a predicted upstream regulator is shown Fig 3E. We used the Ingenuity tool for the analysis of all our gene expression data, as this tool is regularly used in the lab of co-author MI.

5) Finally, the discussion could be improved by adding some recent related literature (Ferris et al., PNAS, 2017; Melo et al., Cell Reports, 2020; Kim et al., Neurobiol Dis, 2020). Findings on brain insulin resistance and metabolic dysfunction could also be discussed (Biessels & Reagan, Nat Rev Neurosci, 2015; Clarke et al., J Alz Dis, 2018).

As suggested by the reviewer, we widened the discussion and added the very relevant literature as proposed by the reviewer.

6) Statistics: Given there are multiple conditions in this experimental design, it is important to ensure proper statistical analyses were done. The authors should indicate whether one-way or two-way ANOVA was conducted, whether they are two-sided, which post-hoc tests were used, etc.

Thank you for notifying us of this missing information. We have added the information on ANOVA (one-way) to the Methods section and information to the respective post-hoc tests to the figure legends.

Reviewer: 2

Comments to the Author(s)

In this study, the authors aimed to investigate the effects of ectopic lipid accumulation in human astrocytes in the insulin signaling in vitro. To do so, Heni and colleagues cultivated human astrocytes (Lonza) in the presence of a lipid emulsion (Lipofundin®) for seven days and executed cell culture, lipid stain, gene expression, and western blotting experiments. Chronic exposure to lipid emulsion increased histological visualization of ectopic fat storage in astrocytes after three days, which remains until the seventh day of treatment and for another three days without Lipofundin®.

To understand how lipid exposure affected gene expression in astrocytes, the transcriptome profiling was analyzed in response to insulin stimuli (3 hours) in the presence of Lipofundin® or after Lipofundin® was withdrawn from the culture medium. Results showed that gene expression was affected diversely in response to lipids and/or insulin, being reversible or not with lipid removal. In this line, Heni and colleagues highlighted the inverse expression profile of some genes in response to insulin stimuli in the presence or absence of lipid emulsion. Moreover, some genes were regulated by insulin regardless of the presence of lipids. Among such genes, several were regulated by the transcription factors FOXO 1 and 3, a known target of the insulin signaling pathway. Thus, to evaluate astrocytic insulin signaling, the authors investigated insulin-induced phosphorylation of Akt, glycogen synthesis, and cell proliferation after lipid exposure (four days) or three days after removal of Lipofundin® from the cell culture medium. The authors suggested insulin signaling impairment as a result of ectopic lipid accumulation.

Although the effect of excessive lipid exposure and the formation of astrocytic lipid droplets is not a new concept, the analysis of the expression profile in response to the presence or absence of the lipid emulsion suggested possible reversibility of lipid toxicity in the astrocytic insulin signaling dysfunction. The manuscript focused on correlating lipid accumulation with insulin signaling dysfunction, but some points still need to be reinforced:

1) The human astrocyte culture needs to be better characterized. How is the expression of classical in vitro cultured astrocyte markers (ALDH1L1, EAAT1, EAAT2, GFAP)? For example, what is the percentage of GFAP positive cells? What was the number of cells used in the experiments? Doesn't the use of human cells, even if commercially obtained, require any type of approval for experimentation?

We thank the reviewer for this important note and provide the reviewer with direct information which we gathered from Lonza (attached and the end of this document). Quote from the document provided by Lonza for the Acquisition of Human Tissue for Research Cell Products: "For research cell products provided by Lonza, we hold donor consent and legal authorization that provides permission for all research use. The consent and authorization documents for Lonza's research cell products do not identify specific types of research testing that can or cannot be performed. If the customer is using Lonza's cells for research purposes only, this donor consent applies. The researcher is responsible for the testing and type of research performed on the cells."

As part of their quality control concept, Lonza performs routine characterization of normal human astrocytes that includes immunofluorescence staining and morphological observation throughout serial passages and staining for glial fibrillary acid protein (GFAP). Only batches with > 80% cells positive for GFAP are considered for sale.

2) It was not clear whether the treatment with the lipid emulsion was done. Lipofundin® was used only once or repeated with each change of cell medium?

We apologize that the procedure was not clearly enough described. Medium with Lipofundin was always freshly prepared and therefore with every change of medium Lipofundin was also refreshed. It was kept in the wells for the indicated time points. We added this information in the methods section.

3) The cell viability test using lactate dehydrogenase activity was analyzed after five days of treatment. How is the viability of the human astrocyte culture after seven days of Lipofundin® treatment and after Lipofundin® was withdrawn from the cell culture medium?

Unfortunately influence of Lipofundin incubation was tested only for five days. Visual inspection of the cells did not indicate severe damage. Though, we address this limitation now in the limitations paragraph in the discussion.

Although lactate dehydrogenase (LDH) released into the medium was measured as a marker of cell death, it is a cell viability measure. If authors would like to express cell death, maybe consider the use of another complementary analysis, for example live/dead assay.

Thank you for notifying us of this imprecise wording. We now explicitly state that this is a cell viability measure.

How was the LDH release calculated about basal conditions? One suggestion is to express the data as dot-plot and insert the error bar in the control (basal) condition to represent the variability between different cultures. Moreover, is not clear how statistics analysis was performed. Which post-hoc test was used, and which comparisons were made?

As suggested by the reviewer, we now present data as box plots with individual data points. We now also added additional information on the statistics and post-hoc test to the manuscript (Methods section and respective figure legends).

4) How was the Lipofundin® withdrawal protocol? Was there any washing before change cell culture media?

Lipofundin was withdrawn from the cells by washing the plates twice (with Lipofundin-free medium) and adding new medium. This is now explained in the methods section.

5) In figure 2, more information is needed on: microscope and objective used, scale bar, how many cultures were used and how many images for each culture was analyzed. Is the same cell culture seen over time or different cell cultures in each time point?

We added this information in the methods section and to the figure.

In the "no more Lipofundin" images, check the number of days on the panels.

Thank you, we corrected the numbers.

6) In figure 3, the heat-maps have a nice big picture of the effects observed in different experimental

conditions. Perhaps, a graphical (maybe a volcano plot?) showing the variation of the genes related to insulin signaling (regulated by FOXO) would highlight the focus of the manuscript discussion and statistical analysis used to include genes.

We thank the reviewer for this very helpful suggestion and added an additional figure (Fig 3E). This figure is a result of the upstream regulator analysis where FOXO was predicted to be inhibited and this is in accordance with the FOXO target genes in this figure that were downregulated.

7) In figure 4, how did the levels of Akt phosphorylation be normalized? How are the total Akt levels in the different conditions compared to the baseline? Was there a loading control for these blottings? Which post-hoc test was done and how were the comparisons made between the different conditions? One suggestion is to express the data as dot-plot and insert the error bar in the control (basal) condition to represent the variability between different cultures.

As suggested by the reviewer, we now present data as box plots with individual data points. We now also added additional information on the statistics and post-hoc test to the manuscript (Methods section and respective figure legends). We now describe in more detail the calculation of the displayed values in the figure legend. The signal intensity of the antibody against phosphorylated serine 473 of Akt was normalized to the signal intensity of the antibody against total Akt and the control condition, which was left untreated, was set as 100 %.

8) Considering the interest in the correlation between the ectopic lipid accumulation and insulin signaling impairment, it would be interesting to assess the total and phosphorylation levels of the insulin receptor.

We agree with the reviewer. Indeed, we tried to quantify this. However, the limited protein amounts available did not allow reliable detection of insulin receptor phosphorylation in our western blots. This has now been added to the limitations part of the manuscript.

9) As a general suggestion, more technical information could be added to the methodology section, which would facilitate the understanding and reproducibility of the experiments. In addition, the figure legends could be more objective and bring more information about the panel data.

Thanks for this comment. We now added further information to the text and the figure legends.

Lonza Walkersville, Inc.
8830 Biggs Ford Road
Walkersville, MD 21793
Tel +1 301 898 7025

Acquisition of Human Tissue for Research Cell Products

Ethics and Law

Established ethical practices of the donation and transplantation organizations in the US (AATB*, AOPO*, EBAA*) are followed at Lonza. Informed consent, legal authorization and protection of human subjects considerations are followed during all steps of the tissue acquisition process. Protected health information is maintained confidentially to protect the privacy of donors as intended by HIPAA* regulation. Lonza holds a permit to operate a tissue bank in the state of Maryland and a tissue bank license from the state of New York for non-transplant anatomic tissue.

Tissue Sources

Human tissue is acquired from tissue recovery agencies, tissue suppliers and Lonza managed donor programs that perform tissue recovery and donor informed consent in accordance with processes approved by an Institutional Review Board or comparable independent review body, where applicable. Good business practices are followed to identify, evaluate, qualify, monitor and maintain agreements with tissue sources. Each qualified tissue source works with Lonza under specific terms defined in a written agreement that includes donor confidentiality, tissue ownership and use, and informed consent or donation permission.

Permission for Research Use

A properly executed record of informed consent from living donors, or authorization for donation from the donor or authorizing person for deceased donors, is required for each human tissue received at Lonza. The intended use of the tissue is included in the document. When a qualified tissue supplier obtains permission for tissue donation either by consent or authorization, that agency maintains the original record and may provide a copy or an attestation statement to Lonza. When the Lonza Donor Program obtains informed consent, the original signed record is retained by Lonza. Informed consent records contain the required elements of informed consent as stipulated in the US regulations for the protection of human subjects 45 CFR* Part 46 and 21 CFR Parts 50 & 56. Authorization records include appropriate language and process as intended in the US Uniform Anatomical Gift Act (UAGA*) (1987) and in updated versions (2006) as enacted by different states.

Non-payment for Human Tissue

Neither living donors, nor family members, nor estates of deceased donors receive valuable consideration for providing tissue. Fees are paid to tissue sources as reimbursement for the medical, technical and transportation services and supplies needed to recover and provide tissue to Lonza. For Lonza Donor Programs, no payment is given to donors whose involvement in providing tissue for research is limited to informed consent. Payment is provided to living donors when additional services are necessary, such as for tissue donor screening, blood sample collection or tissue recovery that may include time and travel.

Donor Anonymity

Lonza ensures that each donor can be identified to a particular tissue and to all cell products derived from that tissue to fulfill tracking requirements. To protect donor identity, tissues and cell products derived from such tissues are coded. Access to donor records and coding is restricted. Product identification codes can be linked to specific attributes of the donor, such as gender, age, medically relevant information or infectious disease test results to provide that information to the customer.

Lonza

Research Use Only

Customers receive a Certificate of Analysis (COA) with each cell product derived from human tissue. The COA contains specific technical information and quality control results pertinent to that product. In addition, the following statements are applicable to all of Lonza's research cell products:

These cells were isolated from donated human tissue after obtaining permission for their use in research applications by informed consent or legal authorization. This product is for research use only. Details concerning the use of our cell and media products can be downloaded from our website at www.lonza.com

For research cell products provided by Lonza, we hold donor consent and legal authorization that provides permission for all research use. The consent and authorization documents for Lonza's research cell products do not identify specific types of research testing that can or cannot be performed. If the customer is using Lonza's cells for research purposes only, this donor consent applies. The researcher is responsible for the testing and type of research performed on the cells.

Authorization

To protect privacy of donors, tissue suppliers, and our proprietary processes, Lonza does not provide copies of donor records or tissue source agreements to customers. This document and/or quality audit practices are offered to customers to provide additional assurance of process integrity.

Quality Program Approval

Michele Jones, Director Quality Assurance
Lonza Walkersville, Inc.

Abbreviations*

AA TB: American Association of Tissue Banks
AOPO: Association of Organ Procurement Organizations
EBAA: Eye Bank Association of America
HIPAA: Health Insurance Portability and Accountability Act of 1996
CFR: Code of Federal Regulations
UAGA: Uniform Anatomical Gift Act (1987, revised 2006)

Electronic Signatures

User	Date	Justification
Dock Nancy ndock	07-Sep-2017 21:47:42	Workflow Signoff Approval
Testin Scott stestin	11-Sep-2017 16:23:56	Workflow Signoff Approval
Leclair Laura lleclair	12-Sep-2017 18:49:17	Workflow Signoff Last Approval

Appendix B

Response to Reviewers

We thank you for pointing out aspects to improve our manuscript further.

Please find below a point-to-point response. Changes in the manuscript are marked as track changes.

Reviewer: 2

Comments to the Author(s)

In this manuscript, Heni et al. showed that chronic exposure to lipids induced ectopic fat storage in cultured human astrocytes that result in a reduction in insulin signaling response and altered gene expression profile in these cells. The subject addressed by authors is important because understanding how changes in metabolism can affect astrocyte function has been shown to be relevant in several areas and diseases, as obesity, diabetes, and neurodegenerative disorders. The authors made several improvements regarding mainly the methodological and statistical description of the results, as well as the organization of the manuscript, highlighting the limitations related to the differences in treatment times and analyses performed.

In this line, I still suggest some other improvements:

1. It would be interesting to insert a sentence related to quality control and ethical approval for the use of commercially obtained cells for the analyzes described in this study in the “Cell Culture” section (Materials and Methods).

We now provide information regarding quality control and ethical approval for the use of astrocytes in the Materials and Methods section as suggested by the reviewer.

2. Data now is present as boxplots with individual data points. The graphic representation of the data as a bar graph or boxplot can be chosen by the authors. The previous suggestion was just to insert the data as individual points to allow the visualization of the variation between the different cultures analyzed. Depending on the choice of the type of graph, it is necessary to revise the legends and adapt the description.

We apologize for the unmatched figure legends after we introduced the boxplots and carefully revised the figure legends in the revised version of the manuscript.

3. Considering that the images shown in figure 2B-C are representative, it is relevant to describe the number of cultures in which the ectopic accumulation of lipids was observed or even if it was a single observation. I also suggest inserting the type of microscope and magnification used to image acquisition in the materials and methods section or figure legend.

We sincerely apologize for not including this information in the first revision and included this important information in the revised version of the manuscript.

4. The authors improved the description of the statistical analyzes used in this manuscript. However, in figures 1 and 2 is not clear why each panel/figure used a different ANOVA post-hoc test. Moreover, the lines used to indicate statistical comparison between groups in the graphs are a little confused, mainly when more than one type of statistical analysis was used between different experimental conditions (for example, Figure 2). I suggest inserting a sentence explaining the choice of different post-hoc tests and the adjustment of the lines that indicate the comparisons in the graphs.

Fig. 1 and 2. As suggested by the reviewer, we harmonized statistics and now report p values from unpaired t-tests for all post hoc statistics. Furthermore, we now removed all non-significant markings to make the figures less busy. We thank the reviewer for this comment.